
# Two mechanisms of stratospheric ozone loss in the Northern hemisphere, studied using data assimilation of Odin/SMR atmospheric observations

Kazutoshi Sagi[1,4], Kristell Pérot[1], Donal Murtagh[1], and Yvan Orsolini[2,3]

[1]Department of Earth and Space Sciences, Chalmers University of Technology
[2]Norwegian Institute for Air Research (NILU)
[3]Birkeland Centre for Space Science, University of Bergen
[4]National Institute of Information and Communications Technology (NICT)

*Correspondence to:* Kazutoshi Sagi (sagi@nict.go.jp)

**Abstract.**

Observations from the Odin/Sub-Millimetre Radiometer (SMR) instrument have been assimilated into the DIAMOND model (Dynamic Isentropic Assimilation Model for OdiN Data), in order to estimate the chemical ozone ($O_3$) loss in the stratosphere. This data assimilation technique is described in Sagi and Murtagh (2016), in which it was used to study the inter-annual variability in ozone depletion during the entire Odin operational time and in both hemispheres. Our study focuses on the Arctic region, where two $O_3$ destruction mechanisms play an important role, involving halogen and nitrogen oxides ($NO_x$) chemical families, respectively. The temporal evolution and geographical distribution of $O_3$ loss in the low and middle stratosphere have been investigated between 2002 and 2013. For the first time, this has been done based on the study of a series of winter-spring seasons over more than a decade, spanning very different dynamical conditions. The chemical mechanisms involved in $O_3$ depletion are very sensitive to thermal conditions and dynamical activity, which are extremely variable in the Arctic stratosphere. We have focused our analysis on particularly cold and warm winters, in order to study the influence it has on ozone loss. The winter 2010/2011 is considered as an example for cold conditions. This case, that has been the subject of many studies, was characterised by a very stable vortex associated with particularly low temperatures, which led to an important halogen-induced $O_3$ loss occurring inside the vortex in the lower stratosphere. We found a loss of 2.1 ppmv at an altitude of 450 K in the end of March 2011, which corresponds to the largest ozone depletion in the northern hemisphere observed during the last decade. This result is consistent with other studies. A similar situation was observed during the winters 2004/2005 and 2007/2008, although the amplitude of the $O_3$ destruction was lower. To study the opposite situation, corresponding to a warm and unstable winter in the stratosphere, we performed a composite calculation of four selected cases, 2003/2004, 2005/2006, 2008/2009 and 2012/2013, which were all affected by a major mid-winter sudden stratospheric warming event, related to particularly high dynamical activity. We have shown that such conditions were associated with low $O_3$ loss below 500 K, while $O_3$ depletion in the middle stratosphere, where the role of $NO_x$-induced destruction processes is prevailing, was particularly important. This can mainly be explained by the horizontal mixing of $NO_x$-rich air from lower latitudes with vortex air, that takes place in case of strongly disturbed dynamical situation. In this manuscript, we show that the relative contribution of $O_3$





depletion mechanisms occurring in the lower or in the middle stratosphere is dramatically influenced by dynamical and thermal conditions. We provide confirmation that the $O_3$ loss driven by nitrogen oxides and triggered by stratospheric warmings can outweigh the effects of halogens in the case of a dynamically unstable Arctic winter. This is the first time that such a study has been performed over a long period of time, covering more than ten years of observations.

## 1 Introduction

Stratospheric ozone ($O_3$) protects life on Earth from harmful ultraviolet solar radiation, and plays a key role in the climate system. The release of halogen compounds by human activities led to a global decrease of stratospheric ozone during the second half of the 20th century. Awareness of the threat resulting from this anthropogenic ozone destruction was raised by the discovery of the Antarctic ozone hole (Farman et al., 1985). Several studies, based on long-term satellite measurements, have shown that global ozone is recovering since the end of the nineties (e.g., Jones et al., 2009; Tummon et al., 2015), as a result of the Montreal Protocol (1987) on the control of ozone depleting substances (ODSs).

Ozone depletion observed in the polar lower stratosphere in both hemispheres can be explained by halogen-induced $O_3$ destruction processes. The main chemical species involved are reactive gases containing chlorine and bromine, which are, to a large extent, emitted by human activities. A complex physico-chemical mechanism, including heterogeneous activation of chlorine by reactions on polar stratospheric clouds (PSCs) followed by catalytic $O_3$ destruction, occurs every year in early spring when the sun returns over the high-latitude region (Brasseur and Solomon, 2005). PSC formation is favoured by cold stratospheric temperatures. As a consequence, higher $O_3$ losses are observed in the lower polar stratosphere in such conditions (Kuttippurath et al., 2012).

Stratospheric ozone is also affected by natural chemical processes. In the middle and upper stratosphere, $O_3$ chemistry is driven by different chemical cycles, involving mainly nitrogen oxides ($NO_x$) (e.g. Kuttippurath et al., 2010; Randall, 2005). This $NO_x$-induced $O_3$ depletion also starts in spring, when the vortex fades away and $NO_x$-rich air masses from lower latitudes can enter the polar region. The main source of stratospheric $NO_x$ is the production of NO through reaction of nitrous oxide ($N_2O$) with an excited oxygen atom $O(^1D)$, which occurs at low and middle latitudes around 30 km (Brasseur and Solomon, 2005). Another noteworthy but smaller source of stratospheric $NO_x$ exists at high latitudes in the mesosphere and lower thermosphere, due to the production of NO by energetic particle precipitation (EPP) (Barth, 2003). In winter polar night conditions, NO has a lifetime long enough to be transported down to the stratosphere by the meridional circulation without being photochemically destroyed (Brasseur and Solomon, 2005; Pérot et al., 2014). As it descends in the polar region, NO is partly converted into $NO_2$.

The Arctic winter stratosphere is characterised by a higher dynamical variability than the Antarctic winter stratosphere. Some winters can therefore experience extremely cold conditions while other winters can be affected by sudden stratospheric warmings (SSW). SSW events correspond to a rapid temperature increase of several tens of kelvin over a few days in the high latitude stratosphere (Charlton and Polvani, 2007). They are triggered by planetary waves propagating upward from





the troposphere, disturbing the polar vortex when they break at stratospheric altitudes. These dynamical conditions strongly influence the formation of PSCs, and the relative contributions of the halogen- and $NO_x$ induced cycles described above.

The chemical ozone loss in the stratosphere can be quantified using different methods, e.g. chemical data assimilation (DA), vortex average descent technique, tracer correlation, the match technique, passive subtraction, and Lagrangian transport calculations. Each method has its own strengths and weaknesses. They are described and compared in the WMO report 2007 and in the references herein. Our study is based on the chemical data assimilation technique, which will be explained in Sect. 2.2. Most of the previous studies on $O_3$ loss were focused on the catalytic halogen reaction cycles taking place in the lower stratosphere, since the ozone hole due to anthropogenic emissions of ODSs was a matter of great concern after its discovery. While the ozone destruction processes involving nitrogen oxides had been mentioned prior to that discovery, their year-to-year contribution to $O_3$ loss has not been studied as thoroughly. Konopka et al. (2007) showed that during the winter 2002/03, as the stratosphere was disturbed by a SSW event, ozone depletion driven by nitrogen oxides did outweigh ozone depletion driven by halogens in the polar region in terms of total $O_3$. However, their study was based on the comparison with only one other winter, the cold and quiet Arctic winter 1999/2000. Kuttippurath et al. (2010) studied the contribution of various chemical cycles playing a role in $O_3$ depletion at different altitudes in the polar stratosphere, over the winters 2004/05 to 2009/10, but the $O_3$ loss observed after the breakdown of the vortex during the years affected by a major mid-winter SSW was not their focus. Other studies examined the relative contributions of nitrogen oxides and halogens for specific winters. Using a data assimilation approach, Jackson and Orsolini (2008) identified a second maximum in vortex-mean ozone loss for the winter 2004/05 period near 650 K (approximately 25 km) likely due to the $NO_x$ catalytic cycle, and much stronger loss outside the vortex. Søvde et al. (2011) studied the relative roles of $NO_x$ and halogen-driven $O_3$ loss during the winter 2006/07 using data assimilation and a chemical transport model, and found that $NO_x$-induced loss at 20 hPa was nearly as high as the halogen loss below.

In Sagi and Murtagh (2016), the year-to-year variability in $O_3$ loss between 2001 and 2013 is characterised for the first time over such a long period using a data assimilation approach, for both hemispheres. Our paper is focused on the study of the two main pathways driving the ozone chemical destruction in the stratosphere, as described above. The purpose is to estimate the relative year-to-year contribution of each of these mechanisms, over a period covering more than a decade, with a focus on the Northern hemisphere. Hence, the conclusions of Konopka et al. (2007) will be reassesses and quantified over a much longer period, characterised by a series of major SSW events.

The manuscript is structured as follows. The observations by the Odin/SMR instrument are briefly presented in Section 2, followed by a brief description of the chemical DA method used to estimate the ozone loss. Section 3 gives an overview of the ozone loss observed in the Arctic region, during twelve winters between 2001 and 2013. The mechanisms responsible for the ozone destruction during particularly cold and warm winters are discussed in Sect. 4 and Sect. 5, respectively. Our conclusions are presented in the last section.



## 2 Measurements and method

### 2.1 Odin/SMR

Odin is a Swedish-led satellite, in cooperation with the Canadian, French and Finnish space agencies, launched in 2001 (Murtagh et al., 2002). The satellite follows a sun-synchronous quasi-polar orbit at 580 km, characterised by the nominal

latitude range [82.5°S - 82.5°N] and varying descending/ascending nodes at 06:00-07:00/18:00-19:00 local time, respectively. These parameters are slightly changing with time due to the drifting orbit. The satellite was initially dedicated to aeronomy and astronomy, but has only been used for aeronomy observations since April 2007. The available measurements are then much more frequent after this date. It has also been an European Space Agency (ESA) third party mission since the same year.

The Sub-Millimetre Radiometer (SMR) is one of the instruments aboard Odin. It is a limb emission sounder providing global

vertically resolved measurements of trace gases and temperature from the upper troposphere up to the lower thermosphere. Our study is based on ozone and $N_2O$ measurements.

Stratospheric ozone mixing ratio is retrieved from an emission line centred at 544.6 GHz. These measurements are performed continuously, and the profiles cover the altitude range $17-50$ km with a vertical resolution of $2-3$ km and an estimated single-profile precision of 1.5 ppmv. The data is filtered according to the measurement response, which is the sum of the rows of the

averaging kernel and indicates how much information has been derived from the true state of the atmosphere. In this study, ozone measurements characterised by a response lower than 0.8 are excluded. A detailed comparison study between ozone products retrieved from the measurement of the 544.6 GHz and the 501.8 GHz emission lines is presented in Sagi and Murtagh (2016).

$N_2O$ is commonly used as a tracer for transport in the stratosphere due to its long chemical lifetime. In our study, SMR $N_2O$

observations have been assimilated, in addition to ozone observations, in order to trace stratospheric air motions. $N_2O$ profiles cover the altitude range 12–60 km with an altitude resolution of $\sim$1.5 km. The estimated systematic error is less than 12 ppbv (Urban et al., 2005a). The validation of the $N_2O$ product is reported by Urban et al. (2005b). Other measurement comparisons with the Fourier Transform Spectrometer (FTS) on-board the Atmospheric Chemistry Experiment (ACE) and the Microwave Limb Sounder (MLS) on the Earth Observing System (EOS) Aura satellite are shown by Strong et al. (2008) and Lambert et al.

(2007), respectively.

### 2.2 Estimation of ozone loss using chemical assimilation

We applied the data assimilation (DA) technique using a transport model to estimate the ozone loss as demonstrated earlier (Rösevall et al., 2007b). The DIAMOND (Dynamic Isentropic Assimilation Model for OdiN Data) model is an off-line isentropic transport and assimilation model designed to simulate horizontal ozone transport in the stratosphere with low numerical

diffusion (Rösevall et al., 2008). Horizontal off-line wind-driven advection has been implemented using the Prather transport scheme (Prather, 1986) which is a mass conservative Eulerian scheme. In this study, wind fields obtained from the European Centre for Medium-Range Weather Forecasts (ECMWF) analyses have been used. Isentropic horizontal advection is performed on separate layers with constant potential temperature (PT), between 425 K and 950 K. The first-order upstream scheme was





implemented in the current version of the model in order to take vertical motion into account, namely the diabatic descent occurring inside the polar vortex (Sagi et al., 2014). The diabatic heating rate was derived from SLIMCAT 3d chemical transport model calculations (Chipperfield, 2006). The diabatic heating rates used for this study were available only until 31 April 2013. Profiles of trace species observed by SMR were sequentially assimilated into the advection model. The assimilation scheme

used in the DIAMOND model can be described as  variant of the Kalman filter (Ménard et al., 2000; Ménard and Chang, 2000). More details on the assimilation scheme can be found in (Rösevall et al., 2008).

The chemical ozone loss can be estimated by comparing two ozone fields transported by the DIAMOND model: a passive one and an active one. Passive ozone is transported by winds in the advection model without any chemistry involved, while the active ozone corresponds to the assimilated $O_3$. This field is transported and modified by the increments resulting from the

assimilation of SMR $O_3$ measurements. The difference between the two fields indicates the change resulting from chemical processes that occurred in the atmosphere.

The edge of the polar vortex is generally defined as the maximum gradient of potential vorticity (PV), which is located around the equivalent latitude (EQL) of $65°$(e.g. Nash et al., 1996; Manney et al., 2006; Rex et al., 2006; Grooß and Müller, 2007). However, in the following sections, the daily ozone losses are averaged over the EQL range $70°$N-$90°$N in order to

make sure that only $O_3$ loss occurring inside the polar vortex is taken into account (we refer the reader to Sagi and Murtagh (2016), Sect. 4.1 for more details).

Previous DA-based studies of ozone loss were case studies of the cold winters of 2004/05 (Rösevall et al., 2008; Jackson and Orsolini, 2008; El Amraoui et al., 2008) or 2006/07 (Rösevall et al., 2007a; Søvde et al., 2011). In Sagi and Murtagh (2016), our estimated ozone loss has been compared to Rösevall et al. (2008), Jackson and Orsolini (2008), and El Amraoui et al.

(2008) as well as to the $O_3$ loss derived from a passive tracer method based on SCIAMACHY measurements (Sonkaew et al., 2013). We showed that our estimation is consistent within approximately 0.2 ppmv with the results from those studies.

# 3   Overview of Arctic ozone loss from 2002 to 2013

The temporal evolution of the vortex-mean ozone change in the Northern hemisphere is presented in figure 1 for the twelve winter/spring seasons from 2002 to 2013. The plots correspond to daily zonal means, smoothed using a three-day moving

average. This figure shows that the Arctic $O_3$ loss is extremely variable from one year to another. We consider two different altitude regions. The lower stratosphere, corresponding to the potential temperature range 425 K-500 K, is represented by the grey area, while the mid-stratosphere in the range 600 K-800 K is represented by the red area.

Accumulated ozone losses on 1 April for each winter / spring season in these two ranges are listed in table 1 as well as the maximum losses with the corresponding dates. 1 April has been selected as a reference date for this comparison because

it corresponds to the beginning of the spring season, when the ozone destruction processes are actively ongoing in the high-latitude middle atmosphere.

However, figure 1 and table 1 show that the duration and the date of the maximum loss can be very variable from one year to another. [3] The $O_3$ losses greater than 1 ppmv on 1 April are highlighted in the table. The ratio of the average loss in the lower





stratosphere (425–500 K) to the average loss in the middle stratosphere (600–800 K) on 1 April for each year, which is given as $\Delta O_3^{\mathrm{Lower}}/\Delta O_3^{\mathrm{Middle}}$, is also indicated. A ratio greater than 1 indicates that the halogen-induced $O_3$ depletion below 500 K is more important than the $O_3$ depletion in the middle stratosphere. This value can therefore help us to identify the dominant $O_3$ destruction pathway during a given year.

The largest loss observed in the lower stratosphere occurred in spring 2011, with a maximum in late March. That season was characterised by a particularly cold stratosphere (Sagi and Murtagh, 2016) and the estimation of the associated ozone loss has been the subject of many studies (e.g. Manney et al., 2011; Hurwitz et al., 2011; Sinnhuber et al., 2011; Arnone et al., 2012; Isaksen et al., 2012; Hommel et al., 2014; Khosrawi et al., 2012). The polar vortex was exceptionally strong and the Brewer-Dobson circulation was much weaker than during the other winters, due to an unusually low planetary wave activity in
the troposphere. The air masses inside the vortex remained well isolated from the air outside the vortex. As seen in table 1, the ozone loss was approximately four times more important in the lower stratosphere than in the middle stratosphere on April 1. These specific winter conditions were favourable for the formation of polar stratospheric clouds over a prolonged period of time that induced effective denitrification in the Arctic stratosphere. The $O_3$ destruction was also almost comparable in magnitude to the one in the Antarctic, as approximately 80% of ozone was depleted at the altitude corresponding to the maximum loss.
This particular winter will be discussed more in detail in Section 4. As seen in Fig. 1, an important $O_3$ loss in the lower stratosphere was also observed during other cold winters such as 2004/2005 and 2007/2008 (e.g. Singleton et al., 2007; Jin et al., 2006; Grooß and Müller, 2007; Jackson and Orsolini, 2008; Kuttippurath et al., 2009), which are also characterised by a ratio $\Delta O_3^{\mathrm{Lower}}/\Delta O_3^{\mathrm{Middle}}$ greater than 1 (table 1).

On the other hand, Fig. 1 indicates that, during some other winters, the loss in the mid-stratosphere can be much more
important than the loss observed between 425 and 500 K. In early 2004, 2006, 2009 and 2013 especially, ozone losses in the mid-stratosphere reached at least 1.4 ppmv in volume mixing ratio (VMR), while losses in the lower stratosphere were always below approximately 0.5 ppmv. The loss ratio on 1 April was particularly low (<0.15, see table 1) during these four winters affected by a major mid-winter SSW that led to the breakdown of the polar vortex. These events were followed by the recovery of the vortex, associated with the formation of an elevated stratopause (ES) and a strong descent motion of air
from the mesosphere down to the stratosphere at the end of the winter / beginning of spring (Orsolini et al., 2010; Funke et al., 2014; Bailey et al., 2014). The strong losses observed in the mid-stratosphere during these four winters can be considered as a response to the SSW-induced dynamical perturbations. As we will see in Section 5, this kind of very active dynamical conditions lead to important changes in the meridional distribution of stratospheric species and in the transport between the mesosphere and the stratosphere.

The other years, i.e. 2001/2002, 2002/2003, 2006/2007, 2009/2010 and 2011/2012, show an more even, yet time-varying, combination of lower-stratospheric and mid-stratospheric losses. Some of these winters were affected by a SSW as well, like the winter 2002/2003 studied by Konopka et al. (2007) for example, or the 2009/2010 and 2011/2012 winters. However, in these later cases, the observed reversal of the zonal-mean zonal wind was of shorter duration (less than one week according to the ECMWF wind fields, not shown here), and did not disturb the polar stratosphere as much as the four events mentioned
previously. As we can see in Fig. 1 and table 1, the observed $O_3$ loss in the mid-stratosphere was much lower for these three





winters than for the four cases discussed above. In the two following sections, we will address the case of the particularly cold and warm Arctic winters, in order to further characterise the mechanisms responsible for the chemical ozone destruction in the lower and middle stratosphere.

**Table 1.** Accumulated ozone loss in the Arctic vortex (EQL [70°N - 90°N]) on 1 April (DOY 90 or 91), ratio of the average loss in the lower stratosphere (425–500 K) to the average loss in the middle stratosphere (600–800 K), date of the maximum loss and the corresponding date, for each year between 2002 and 2013. All $O_3$ losses are given in VMR (ppmv). The values lower than -1 ppmv on 1 April are highlighted.

| Year | | 1 Apr. | $\frac{\Delta O_3^{Lower}}{\Delta O_3^{Middle}}$ | Max. | Max. date (DOY) |
|---|---|---|---|---|---|
| 2002 | 425-500 K | -0.42 | 0.48 | -0.93 | 07 Mar. (65) |
| | 600-800 K | -0.87 | | -1.11 | 21 Mar. (79) |
| 2003 | 425-500 K | -0.39 | 0.59 | -0.66 | 11 Mar. (69) |
| | 600-800 K | -0.66 | | -1.06 | 17 Apr. (106) |
| 2004 | 425-500 K | -0.21 | 0.14 | -0.56 | 08 Apr. (98) |
| | 600-800 K | -1.54 | | -1.65 | 04 Apr. (94) |
| 2005 | 425-500 K | -0.28 | 1.67 | -0.68 | 15 Mar. (73) |
| | 600-800 K | -0.17 | | -0.55 | 17 Feb. (47) |
| 2006 | 425-500 K | -0.03 | 0.02 | -0.49 | 04 Mar. (62) |
| | 600-800 K | -1.31 | | -1.41 | 12 Mar. (70) |
| 2007 | 425-500 K | -0.33 | 0.51 | -0.52 | 03 Mar. (61) |
| | 600-800 K | -0.65 | | -0.70 | 03 Apr. (92) |
| 2008 | 425-500 K | -0.60 | 1.15 | -0.84 | 07 Mar. (66) |
| | 600-800 K | -0.52 | | -0.97 | 22 Apr. (112) |
| 2009 | 425-500 K | -0.07 | 0.04 | -0.50 | 10 Feb. (40) |
| | 600-800 K | -1.61 | | -1.82 | 14 Apr. (103) |
| 2010 | 425-500 K | -0.39 | 0.66 | -0.71 | 09 Mar. (67) |
| | 600-800 K | -0.59 | | -0.97 | 22 Mar. (80) |
| 2011 | 425-500 K | -1.30 | 4.22 | -1.47 | 27 Mar. (85) |
| | 600-800 K | -0.31 | | -0.68 | 24 Mar. (82) |
| 2012 | 425-500 K | -0.10 | 0.15 | -0.30 | 08 Feb. (38) |
| | 600-800 K | -0.67 | | -1.11 | 14 Apr. (104) |
| 2013 | 425-500 K | -0.20 | 0.14 | -0.38 | 02 Feb. (32) |
| | 600-800 K | -1.37 | | -1.78 | 27 Mar. (85) |



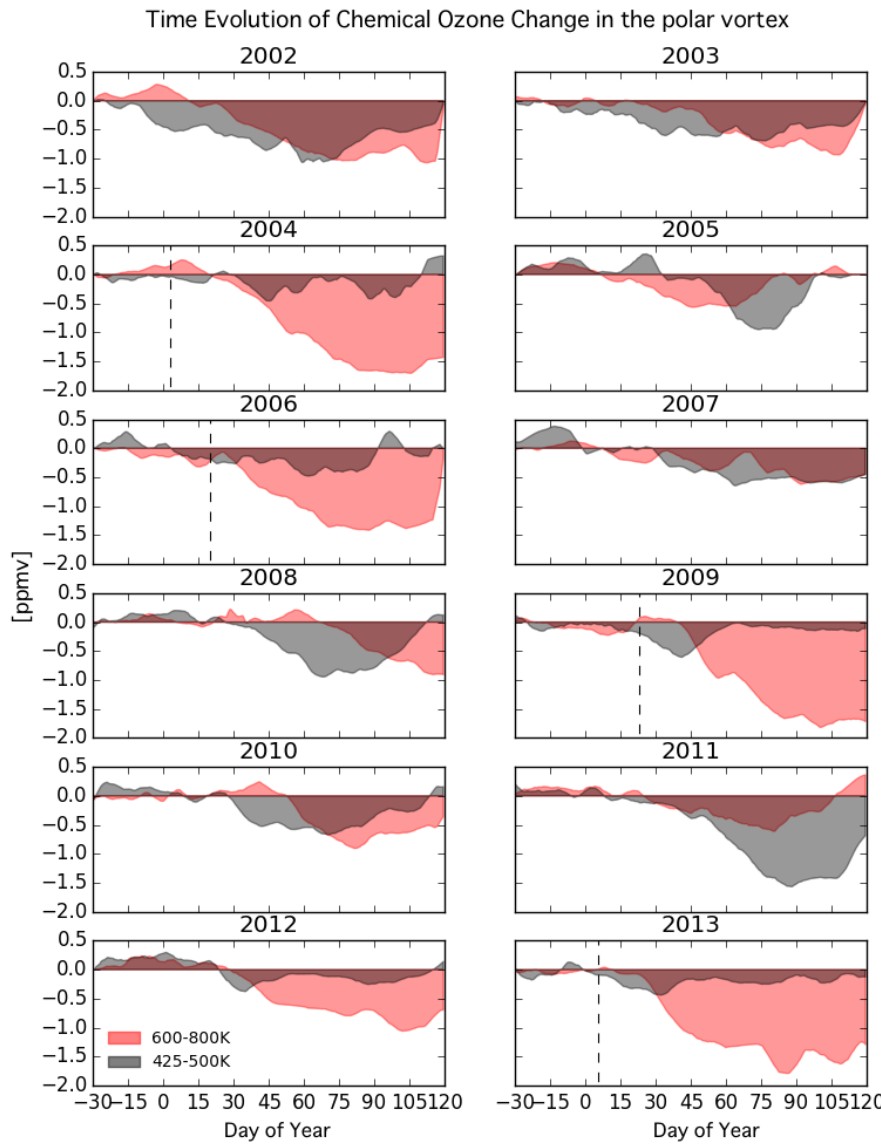

**Figure 1.** Time evolution of the estimated ozone change in the vortex, in volume mixing ratio, for each Arctic winter from 2002 to 2013. The equivalent latitude of $70\,^\circ$N has been used to define the polar vortex border. The gray and red areas show the average of vortex mean ozone change in the lower stratosphere (425–500 K) and in the mid-stratosphere (600–800 K), respectively. The dashed lines in 2004, 2006, 2009 and 2013 indicate the central date of the sudden stratospheric warmings, which is used as a reference date to calculate the composite discussed in section 5.





## 4   Lower stratospheric ozone loss during cold winters

As described in the introduction (Sec. 1), the ozone loss in the polar lower stratosphere can be explained, to a large extent, by chemical destruction processes involving halogen compounds, occurring in spring when cold vortex air is exposed to sun light. The Arctic winter 2010/2011 is a very good example to study these processes. The zonal-mean zonal wind derived from

ECMWF analyses at 55 hPa and 60°N, averaged over the two months of February and March 2011, was around 25 m/s, while the mean for all the years into consideration in our study is 12.5 m/s with a standard deviation of 5.5 m/s. This is the only winter for which the value is above the mean plus one standard deviation, which indicates a particularly strong and stable vortex. This section is hence dedicated to the study of the $O_3$ destruction mechanisms during this specific winter, as an outstanding case of a cold Arctic stratosphere.

Figure 2 shows the temporal evolution of chemical ozone change (left column) and of assimilated $N_2O$ volume mixing ratio (middle column) during the Arctic winter 2010/2011; also shown is the time change of cumulative insolation with an unit of hours per day (right column). The latter was calculated in each model gridbox by looking at the solar zenith angle (SZA), and by transporting this information with the advection model. We used a SZA value of 102°, which is also known as a border of Nautical twilight, as a threshold between day and night. Hence, a local increase of this parameter can indicate both direct

exposure of the air parcel to sun light and mixing with air masses which had received longer insolation.

The top panels of each column represent the vortex mean ([70°N - 90°N] EQL range) for each variable as a function of time and potential temperature. The two panels below show the temporal evolution of the latitudinal distribution in EQL for each variable, averaged over the mid-stratosphere between 600 and 800 K, and the lower stratosphere between 425 K and 500 K, respectively. The white solid lines in these latter plots indicate the equivalent latitude of 70°N, used to delineate the vortex

border. The shaded areas correspond to gaps in the SMR data set. During these periods, the assimilated variables are only transported by the model, with no assimilation increment.

As explained in section 2.1, $N_2O$ is commonly used a tracer for the transport processes in the stratosphere. The latitudinal distribution of $N_2O$ is characterised by a very strong gradient around the EQL of 70°N, which indicates a particularly sharp barrier at the vortex edge. We can see that the chemical $O_3$ loss below 500 K was strictly confined to the interior of the polar

vortex, beginning in mid-February. That time corresponds to the end of the polar night, when the vortex air was exposed to solar radiation and the heterogeneous chemical processes involving halogen compounds could come into effect. The maximum of the vortex averaged ozone loss (Fig.2, upper left pannel) reached 2.1 ppmv around 450 K by the end of March and early April, when more than 80% of the ozone was depleted from the stratosphere at that time and altitude. Figure 3 shows maps of the inferred ozone loss and of assimilated $N_2O$ averaged over the vertical range [425 K - 500 K] on March 27, 2011, when the

maximum $O_3$ loss was observed. The polar vortex border (EQL of 70°N), indicated by the thick solid lines, confines the area of ozone loss, which is perfectly consistent with what we see in figure 2.

These results are consistent with other studies dedicated to this specific winter, although our estimated loss is slightly lower (approximately 0.4 ppmv) (Arnone et al., 2012; Manney et al., 2011; Sinnhuber et al., 2011; Hommel et al., 2014). This is generally the case not only for the Arctic winter into consideration here, but also for other winters and in both hemispheres. Our





ozone loss estimations during the whole time period 2002-2013 has been thoroughly compared to those in other publications in Sagi and Murtagh (2016), and two plausible explanations for the differences are discussed. One reason is the vortex boundary criterion. Another reason is the instrumental quality such as vertical resolution and sensitivity. Considering these issues, we have approximately an uncertainty of 0.2 ppmv in our estimation. The largest difference was found with the $O_3$ loss estimated

by comparing SCIAMACHY ozone measurements and quasi-passive ozone calculated by a chemical transport model (Hommel et al., 2014). The corresponding maximum loss is 1 ppmv higher than our result. Quasi-passive ozone is not completely passive but uses an adapted version of the linearised chemistry scheme excluding heterogeneous chemistry. Thus their quantification only indicates chemical loss by heterogeneous reactions, while ours includes all chemical $O_3$ changes, including $NO_x$ induced and even $O_3$ production.

As we can see in Fig. 2, the spatial distribution of ozone change in the middle stratosphere (600 K–800 K) shows a very different pattern to the distribution in the lower stratosphere. The $O_3$ chemical destruction was observed outside the vortex from the beginning of December and extended through the mid-latitude surf zone up to the vortex edge. A maximum loss of approximately 2 ppmv is observed just south of 70°N of EQL, confined outside of the strong vortex. In contrast, the loss inside the vortex was below 0.6 ppmv, and began in late January 2011, simultaneously with a short enhancement in insolation time

as seen in the right panel. This increase in insolation time denotes the weakening and the deformation of the vortex, which now extends into lower latitudes. The polar maps of $N_2O$ show that the vortex was split into two parts by a minor SSW on 4 February 2011 (not shown here). However, the vortex reformed after only a few days. This explains that a slight $O_3$ destruction was observed in the vortex during the period following this event, induced by horizontal mixing of $NO_x$-rich air.

In summary, an exceptionally important halogen-induced $O_3$ destruction occurred inside the vortex below 500 K in March

and April 2011, while the vortex loss above 600 K was much lower, due to limited mixing of air from lower latitudes. The ozone loss during other relatively cold winters, such as 2004/2005 and 2007/2008, presented similar patterns, although the relative amplitudes of losses in the lower and middle stratosphere varied.

## 5 Mid-stratospheric ozone loss after SSW events

We now focus on the chemical ozone destruction in the mid-stratosphere in warm conditions, during the winters 2003/2004,

2005/2006, 2008/2009 and 2012/2013. A major midwinter stratospheric warming is defined as the sudden reversal of the zonal mean zonal wind at a latitude of 60° and 10 hPa between November and March, associated with a positive zonal-mean temperature gradient between 60° and 90° at the same pressure level (Andrews et al., 1987). In addition, during these four Arctic winters, the reversal of the zonal-mean zonal wind persisted over more than one week according to the ECMWF analyses, which increased the potential of these SSWs to affect the circulation in the middle atmosphere. As already mentioned

in section 3, these events were followed by the recovery of the vortex associated with the formation of an elevated stratopause (Orsolini et al., 2010; Pérot et al., 2014). The SSW central date is defined as the first day of the zonal-mean zonal wind reversal at 10 hPa, and has been chosen as a reference date to calculate the composite of these four winters. It corresponds to 4 January 2004, 21 January 2006, 24 January 2009 and 6 January 2013, respectively.


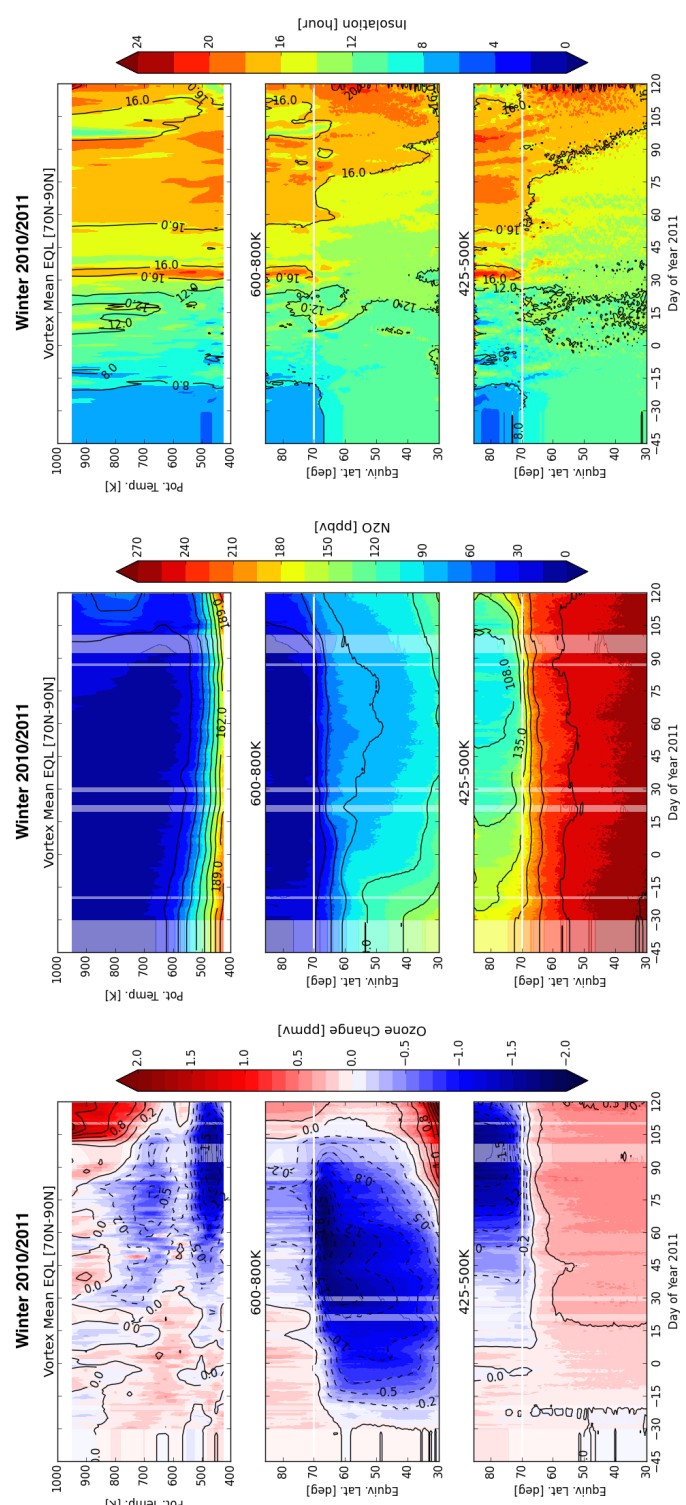

**Figure 2.** Chemical ozone change (left column), assimilated N$_2$O volume mixing ratio (middle column) and the time change of cumulative insolation with an unit of hours per day (right column) during the Arctic winter 2010/2011. The top panels of each column represent the vortex mean for each variable ([70°N−90°N] EQL range) as a function of time and potential temperature. The two panels below show the temporal evolution of the spatial distribution in EQL at selected isentropic surfaces (the mid-stratospheric average between 600 and 800 K and the lower stratospheric average between 425 K and 500 K, respectively). The horizontal white solid lines in these plots indicate the EQL of 70°N, used as the vortex edge border in our study. The shaded areas indicate the gaps in the SMR data set. The values showed during these periods correspond to the transport model only.



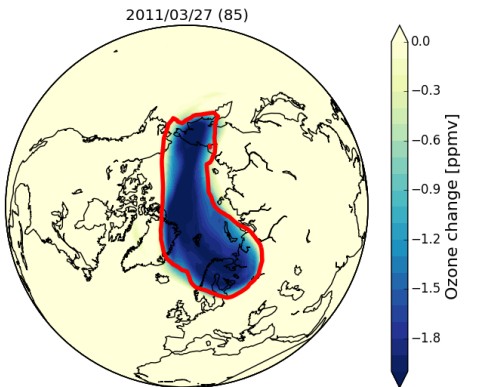
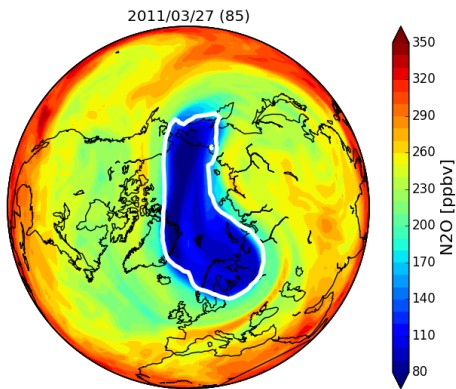

**Figure 3.** Example of maps of the estimated ozone loss (left panel) and of N$_2$O assimilated fields (right panel). Both maps show the average volume mixing ratio in the lower stratosphere (vertical range $[425\,K-500\,K]$) on March 27, 2011, which corresponds to the time and place where the maximum ozone loss occurred. Red and white contour lines indicate the polar vortex border (EQL of $70°N$).

Figure 4 represents the temporal evolution of the chemical ozone change (left column), assimilated N$_2$O volume mixing ratio (middle column) and the time change of cumulative insolation (right column) for the composite of warm winters. These winters were characterised by a temperature increase in the lower stratosphere occurring much earlier than the climatological springtime temperature increase associated with the final warming (Sagi and Murtagh, 2016). These warm conditions were not

favourable to PSC formation, which explains that the ozone loss in the vertical range 425-500 K is particularly low during these years, as seen by the contrast with Fig. 2. This loss started from the SSW central date and was confined to the interior of the vortex. The maximum loss in the composite in the lower stratosphere is 0.5 ppmv at 500 K around 15 days after the central date, corresponding to the vortex distortion due to the warming event. This signature is consistent with the loss observed during the Arctic winter 2012/13 using the Aura/MLS instrument by (Manney et al., 2015), who explained that moderately cold conditions

in December 2012 resulted in extensive PSC formation before the SSW. A combination of early chlorine activation on PSCs and slow chlorine deactivation due to denitrification led to O$_3$ loss in January 2013, following the warming event.

These warm winters affected by strong dynamical perturbations are however characterised by important ozone loss in the mid-stratosphere. As we can see in figure 4 in the PT range 600 K–800 K, O$_3$ is depleted outside the vortex starting already in December at low EQL. This ozone depletion expands over a wider range of EQL over the course of the winter, and is not

bounded by EQL of $70°$. It reaches the vortex approximately 2 or 3 weeks after the SSW central date. Just before that, we can note an ozone increase (hence production) at high EQL, which coincides with an increase in insolation time of the vortex air (Fig. 4, right panels) due to its displacement towards lower geographic latitudes resulting from the SSW. Solar exposure owing to these dynamical perturbations triggers changes in local photochemical equilibrium and leads to ozone production. This is in contrast with PT levels below 500 K, where the vortex air exposure to sun light induced O$_3$ destruction due to heterogeneous

activation of chlorine. After the brief production period, a particularly strong chemical ozone loss is observed at high equivalent




latitudes, extending from 600 K to near 900 K, as there is mixing between vortex air and $NO_x$-rich air from lower latitudes over a broad altitude range. This horizontal mixing is visible in the middle panel of Fig. 4, representing the temporal evolution of the $N_2O$ spatial distribution. This loss is stronger higher up in late January and descends down to 600 K, where it significantly increases up to its maximum around 90 days after the central date. The beginning of the period corresponding to an ozone loss

higher than 1.5 ppmv coincides with the vortex recovery. As explained in section 1, downward transport of $NO_x$ produced by energetic particle precipitation in the MLT during winter is another source of stratospheric $NO_x$. This is especially true in the case of a winter affected by a SSW-ES event, when this descent motion starts higher than usual and can thus bring more NO down from the mesosphere (Orsolini et al., 2010; Pérot et al., 2014). We therefore expect to see an impact of EPP-$NO_x$ on ozone in the middle stratosphere sometime after the warming event. However, we were not able to distinguish this effect from

the impact of the horizontal mixing of air masses in the framework of our study, because it was not possible to assimilate SMR NO observations.

In order to describe in more detail the $NO_x$-induced ozone loss in the case of a warm winter, we look now specifically at $O_3$ loss and $N_2O$ polar maps averaged between 600 K and 800 K for four selected days after the onset of the SSW in Arctic winter 2012/2013 (figure 5). At the beginning, the vortex was elongated and displaced from the pole, and the $O_3$ chemical loss was

observed only outside the vortex (left column). Soon after the central date, the vortex was split into several smaller vortices (second column), and the vortex air was mixed with air parcels from lower latitudes. After the recovery, the $O_3$ destruction occurred mainly inside the vortex (third column). The chemical loss progressively increased and moved towards the pole as the $NO_x$-rich air was transported into the vortex, with a maximum at the end of March, as seen in the right column. At that time, there could also be a possible effect of the EPP-$NO_x$ transported downwards in the vortex. The vortex was sustained even after

the end of the polar night, which is consistent with the findings of Thiéblemont et al. (2013) who showed that, when a strong SSW occurs, the final warming tends to occur later than in other years. The inferred ozone loss was at least 2 ppmv by the time of the final warming.

The composite represented in Fig. 4 shows a good similarity in the vertical and horizontal distribution of chemical ozone change with the case study of the winter 2002/2003 discussed in Konopka et al. (2007). As explained in the introduction (section

1), $NO_x$-induced chemical reactions leading to $O_3$ depletion play an important role in the altitude range into consideration here. The study of the composite of these four winters show that this $O_3$ loss mechanism becomes predominant in the case of strong dynamical perturbations due a major mid-winter SSW-ES event.

## 6   Conclusions

We assessed the chemical ozone loss in the Northern hemisphere in order to document the inter-annual variability of halogen-

induced loss occurring in the lower stratosphere in comparison to the loss in the mid-stratosphere, mainly due to chemical reactions involving $NO_x$ species. We applied a data assimilation approach using the off-line wind driven isentropic transport and assimilation model DIAMOND. Ozone vertical profiles retrieved from the emission line at 544 GHz observed by Odin/SMR were assimilated into the DIAMOND model in order to obtain spatial and temporal ozone distributions at potential



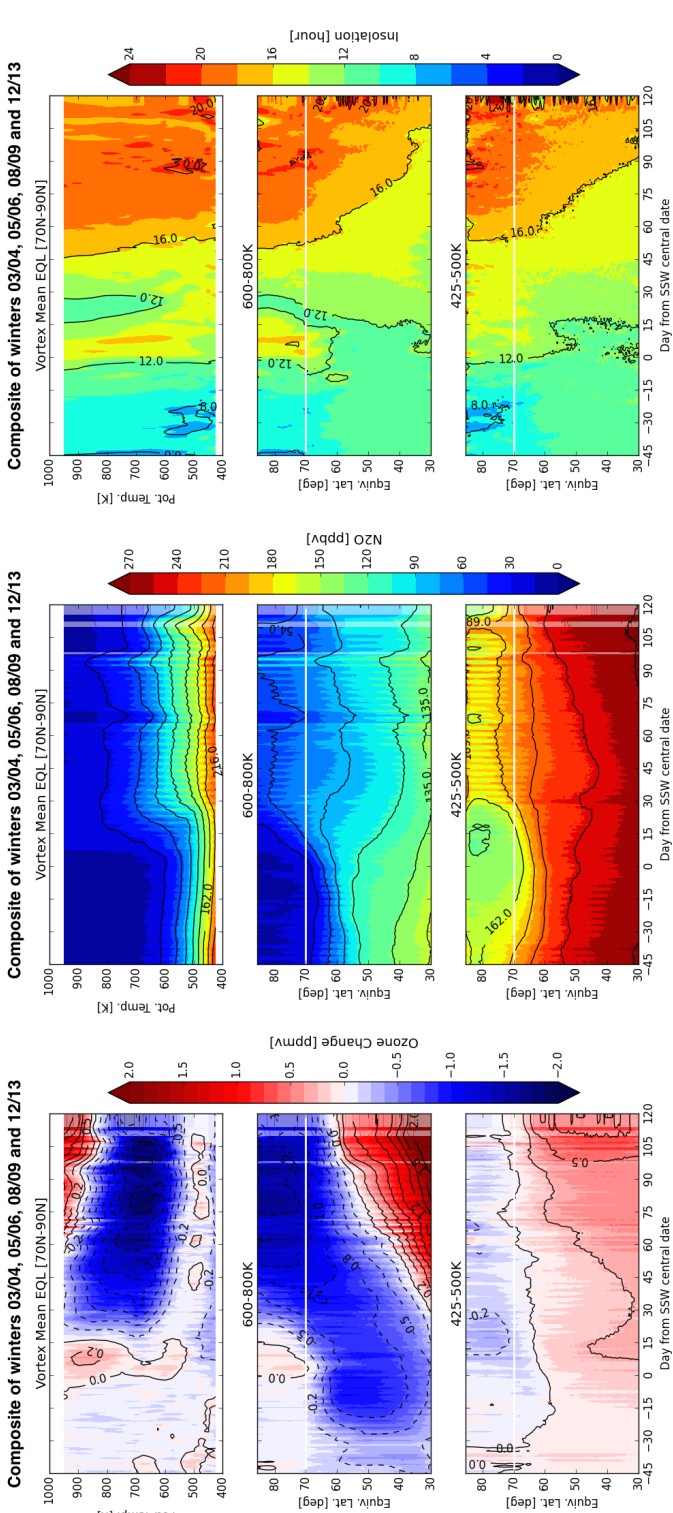

**Figure 4.** Same as figure 2 for the composite of the Arctic winters 2003/2004, 2005/2006, 2008/2009 and 2012/2013, affected by a mid-winter major sudden stratospheric warming. The time is expressed in days relative to SSW central date.




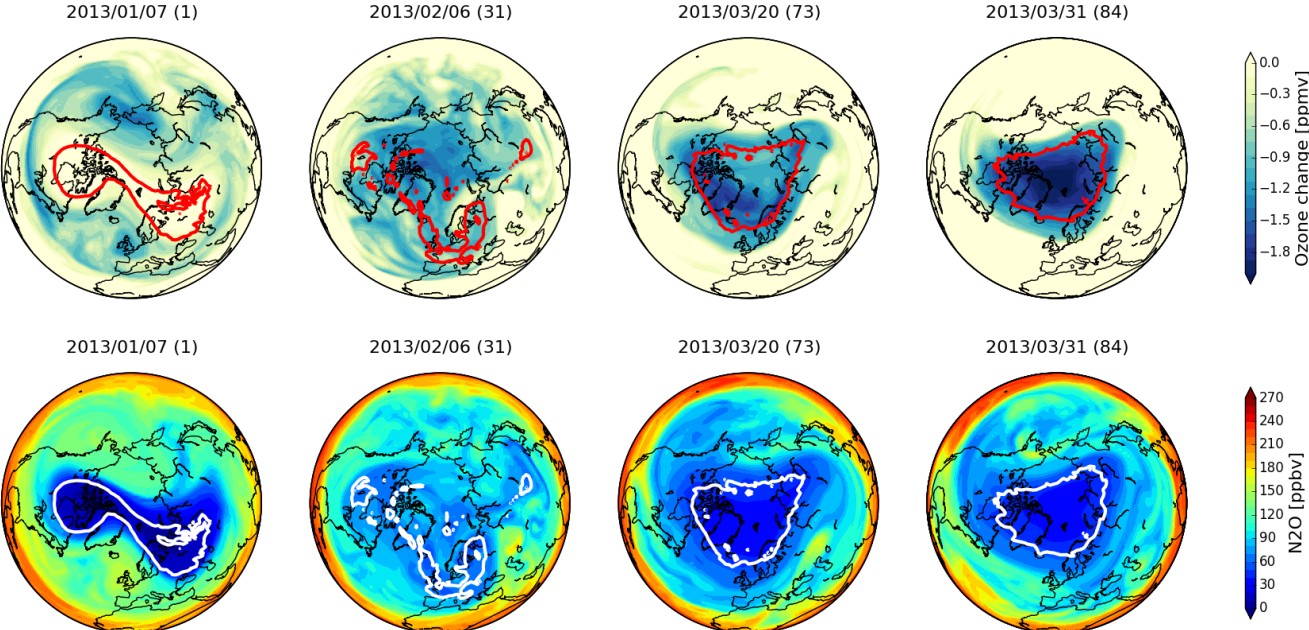

**Figure 5.** Ozone loss (top panels) and N$_2$O (bottom panels) maps in the mid-stratosphere (600–800 K) for four selected dates in early 2013. The number of days after the SSW central date (2013/01/06) is given in brackets. Thus, the first map corresponds to the SSW onset , the second one to the break up of the polar vortex , the third one to the recovery of the vortex and the last map corresponds to the beginning of the spring.

temperatures between 425 K and 950 K in the Northern hemisphere. Assimilation experiments of SMR ozone measurements were performed for each Arctic winter between 2001 and 2013. The analysis period for each assimilation experiment runs from 1 December to 31 April. Inferred chemical ozone loss was calculated by subtraction of passive ozone, which corresponds to passively transported ozone fields by DIAMOND model, from the assimilated (or active) ozone. The Arctic ozone losses both

5 in the lower and mid stratosphere are characterised by a large inter-annual variability.

To describe the ClO$_x$-induced ozone loss in the case of cold winters, we chose to focus on the 2010/2011 winter, when the vortex was exceptionally strong in February and March. The sharp vortex barrier during this winter, seen in Figure 2, allowed a confined ozone depletion to occur in the lower stratosphere below 500 K, with a maximum loss of 2.1 ppmv (0.2 ppmv uncertainty) at 450 K on 27 March 2011. On the other hand, the ozone loss in the middle stratosphere (600 K-800 K) is

10 opposite, remaining low due to limited horizontal mixing into the strong vortex. Similar tendencies are also seen in the other relatively cold winters, such as 2004/05 and 2007/08. Note that during these cold winters, the loss ratio $\Delta O_3^{\mathrm{Lower}}/\Delta O_3^{\mathrm{Middle}}$ was greater than 1 on 1 April (see table 1).

Konopka et al. (2007) showed, in a case study of the 2002/03 winter affected by a SSW, that NO$_x$-induced loss was comparable to or could even outweigh ClO$_x$-induced loss (albeit at different heights), and was mostly due to meridional transport of

15 NO$_x$-rich air from lower latitudes. Here we have re-assessed and quantified these conclusions over a much longer period, span-



ning more than a decade characterised by a series of major SSWs. Pronounced mid-stratospheric ozone losses are consistent with occurrences of such major SSW events, and their attendant large transport from lower latitudes, as revealed in a composite of the four winters 2003/2004, 2005/2006, 2008/2009 and 2012/2013. This loss begins at higher up in late January then descends down to 600 K. The inferred loss of more than 1.5 ppmv between 600–800 K occurs with the vortex recovery in all four winters selected for the composite analysis. During these four events characterised by prolonged zonal wind reversal, the contribution of the $NO_x$-induced loss was even more pronounced –broadly by a factor 2- than during the warming considered in Konopka et al. (2007).

As shown in this article, ozone depletion in both the lower and middle Arctic stratosphere are dramatically influenced by dynamical and thermal conditions. Meanwhile, it is expected that EPP indirectly affects the stratospheric ozone during the polar winter. This is specially true in the southern hemisphere, where Fytterer et al. (2015) indicated contributions of EPP-$NO_x$, related to geomagnetic activity, on the Antarctic ozone depletion between 2005 and 2010. However, according to Konopka et al. (2007), in the 2002/2003 Arctic winter, the $NO_x$ descent from the mesosphere had a minor impact upon the stratospheric ozone depletion in comparison to the meridional transport, as the mesospheric $NO_x$ did not propagate low enough to reach the mid-stratosphere. While the warm winters considered in this study, i.e. 2003/2004, 2005/2006, 2008/2009, and 2012/2013 were characterised by a strong mesospheric descent (Pérot et al., 2014), we could not find clear evidence that it played a significant role in the mid-stratospheric ozone depletion. This is partly due to the insufficient temporal sampling of NO observations by the SMR instrument for an assimilation study. Furthermore, assimilation of $NO_x$ would require the development of a model with relevant stratospheric chemistry. Such investigations would be necessary in order to quantify the small contribution of the downward transport of EPP-$NO_x$ from the contribution of horizontal transport of $NO_x$ from lower latitudes.

*Acknowledgements.* Odin is a Swedish-led satellite project funded jointly by the Swedish National Space Board (SNSB), the Canadian Space Agency (CSA), the National Technology Agency of Finland (Tekes), the Centre National d'Études Spatiales (CNES) in France and the European Space Agency (ESA). Yvan Orsolini is partly supported at the Birkeland Centre for Space Science by the Research Council of Norway/CoE under contract 223252/F50. We thank the study group on the added-value of chemical data assimilation in the stratosphere and upper-troposphere supported by the International Space Science Institute (ISSI). We thank Martyn Chipperfield and Wuhu Feng from the University of Leeds for providing the diabatic heating rates for this study.



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
