# Peer review of "Two mechanisms of stratospheric ozone loss in the Northern hemisphere, studied using data assimilation of Odin/SMR atmospheric observations"

_Atmospheric Chemistry and Physics, 2016_

## Referee Comment (RC1) · Anonymous Referee #1 · 1 Aug 2016

[referee-annotated manuscript omitted]

---

## Referee Comment (RC2) · Anonymous Referee #2 · 4 Aug 2016

**General:**

The paper presents the analysis of stratospheric ozone loss in the Northern hemisphere during the period 2002-13 as derived from the DIAMOND model simulations which are assimilated to the Odin/SMR satellite observations. It compares the chlorine-induced with NOx-induced ozone loss and shows that the second effect is extremely important during the warm NH winters which are disturbed by the SSWs. This important contribution is well-written and is supported by well-performed figures. The paper should be acceptable after minor revision with some minor points listed below.

[Figure]

**Minor points:**

1. Par. 2.2

   You use the DIAMOND model that contains only the isentropic transport. On the other side the cross-isentropic transport is also implemented in form of the diabatic heating rates. In total, you have a full 3d transport that is certainly more than the DIAMOND model. ...so maybe "extended version of the DIAMOND" model describes better your approach.

2. Table 1

   It would be also desirable to list in Table 1 the integrated ozone loss (in Dobson units) for both considered regions. This would quantify the respective contributions to the column ozone loss and is for me a more reliable (or more comparable) quantity than the mean ozone loss

3. Table 1, caption

   ..., value of the maximum loss and the corresponding date

4. P5 L28

   1th April

5. P5 L33

   "3" typo

6. P6 L14

   ..in magnitude to the typical ozone loss in the Antarctic...

7. Figure 2 and 4

   In the last column you show the isentropic view of the cumulative insolation, i.e. between 600 and 800K as well as between 425 and 500K. Maybe you should explain why the highest values start in the polar region (where the polar night is

still expected) and move to the south where the sun activity should be the highest. Is it because of the Earth geometry in the relation to the position of the Sun? This insolation effect seems to be similar for the cold and warm winter. Few additional sentences would help.

8. P12 L13
   PT-range, I would prefer $\theta$-range

9. P13
   I would recommend to write out the abbreviations MLT and ES because they are only sporadically used.

10. P13 L31
    "using the off-line wind driven isentropic transport" - I think your transport is 3d (diabatic heating), and this effect is important for the here considered time periods of the order of few months

11. P15, L5-15
    Here, once again Dobson units defining the contribution to the total column would be better

12. P16, L3
    This loss begins at higher altitudes in late January.

---

## Author Comment (AC1) · 11 Nov 2016

Dear referee,

We greatly acknowledge your helpful comments. All the changes we made based on your suggestions are written in blue in the attached revised manuscript.

Please find our detailed replies to your comments in the attached file as well as the revised manuscript.

Please also note the supplement to this comment:

[Figure]

http://www.atmos-chem-phys-discuss.net/acp-2016-511/acp-2016-511-AC1-supplement.zip

---

## Author Comment (AC2) · 11 Nov 2016

Dear referee,

We greatly acknowledge your helpful comments. All the changes we made based on your suggestions are written in red in the attached revised manuscript.

We have prepared our detailed replies to your comments in the attached file. Please find them in the attached supplement.

Please also note the supplement to this comment:

http://www.atmos-chem-phys-discuss.net/acp-2016-511/acp-2016-511-AC2-supplement.zip

---

## Author Response (AR1)

**Response to anonymous referee #1**

Dear referee,

We greatly acknowledge your helpful comments. All the changes we made based on your suggestions are written in blue in the revised manuscript. Below we give more details about some of the improvements we made, or we explain why we did not change the article as you suggested in some cases:

p.1, l.16: "Identify these other studies here."

➢ We have decided not to add more citations in the abstract, because it would go against a modification we had already made before publication in ACPD, following a suggestion by referee #2 (in his/her quick report). However these other studies are identified in Sect.4 and Sect.5.

p.3,l.8: "Identify ODS acronym."

➢ That is already done on page 2 (l.12).

Sect. 3 and 6:

➢ We have explained more in detail the choice of the four winters included in the composite (p.6-7), as well as the difference between the $NO_x$-induced ozone loss corresponding to this composite and to the winter 2002/03, studied by Konopka et al. (2007) (p.16). The figure below (not included in the manuscript), representing the temporal evolution of the zonal mean zonal wind at 60°N and 10hPa from ECMWF during the twelve winters into consideration in our study, is supporting our explanations.

[Figure]

The new text hence reads : "In other years, such as 2001/2002, 2002/2003, 2006/07 and 2009/10, the maximum losses in the mid and lower stratospheric layers were comparable. In fact, several winters witnessed the occurrence of a mid-winter SSW in addition to the four above-mentioned events (in 2003/2004, 2005/2006, 2008/2009 and 2012/2013). This is the case for the winter 2002/2003 studied by Konopka et al. (2007), as well as the 2009/2010 winter. However, in these two cases, the reversal of the zonal-mean zonal wind at 60N and 10 hPa, as deduced from the ECMWF analyses, was shorter than a week, and the polar stratosphere was less disturbed.  As we can see in Fig. 1 and table 1, the observed O3 loss in the mid-stratosphere was much lower for these less-disturbed winters than for the four above-mentioned events in 2003/2004, 2005/2006, 2008/2009 and 2012/2013. The winter 2011/2012 was characterized by a minor SSW (no reversal of the zonal wind at 60N and 10 hPa was observed), and in that case the mid-stratospheric loss also overwhelmed the lower stratospheric loss, with a loss ratio comparable to the other four above-mentioned events. Since the mid-stratospheric loss was not greater than 1 ppmv in magnitude, this event was not retained in the composite, but this subjective choice does not affect greatly our results."

Also the attached figure supports the new statement in the conclusion section on the comparison with Konopka's study: "During these four events, the contribution of the NOx-induced loss was even more pronounced – broadly by a factor 2 - than during the warming considered in Konopka et al. (2007). The longer duration of the zonal wind reversal, as explained in Section 3 and the resulting larger disruption of the polar stratosphere, can explain this significant difference."

Dear referee,

We greatly acknowledge your helpful comments and suggestions. Below we present the detailed replies to each comment. All modifications are written in red in the revised manuscript.

**Minor points:**
1. Par. 2.2
You use the DIAMOND model that contains only the isentropic transport. On the other side the cross-isentropic transport is also implemented in form of the diabatic heating rates. In total, you have a full 3d transport that is certainly more than the DIAMOND model. ...so maybe "extended version of the DIAMOND" model describes better your approach.
➢      This precision has been added.

2. Table 1
It would be also desirable to list in Table 1 the integrated ozone loss (in Dobson units) for both considered regions. This would quantify the respective contributions to the column ozone loss and is for me a more reliable (or more comparable) quantity than the mean ozone loss
➢      The column ozone change in DU has been added to table 1, as an additional information, in order to make the comparison with other studies easier. However, it is important to keep in mind that the integrated O3 loss values also depends on the size of the vortex area, that can be different from one year to another. This integrated quantity is not suited to study the composite of several winter seasons, as done in Sect. 5. For the sake of consistency, we have decided to keep our discussion based on the vmr quantity in the whole paper.

3. Table 1, caption
..., value of the maximum loss and the corresponding date
➢      This has been corrected.

4. P5 L28
1th April
➢      "1 April" has been changed to "1st April". For the sake of consistency, all dates have been written using the same format.

5. P5 L33
"3" typo
➢      This has been corrected.

6. P6 L14
..in magnitude to the typical ozone loss in the Antarctic...
➢      This has been changed.

7. Figure 2 and 4
In the last column you show the isentropic view of the cumulative insolation, i.e. between 600 and 800K as well as between 425 and 500K. Maybe you should explain why the highest values start in the polar region (where the polar night is still expected) and move to the south where the sun activity should be the highest. Is it because of the Earth geometry in the relation to the position of the Sun?

This insolation effect seems to be similar for the cold and warm winter. Few additional sentences would help.

➢ The last column of Fig.2 and 4 does not represent the cumulative insolation, but the time change of cumulative insolation, expressed in number of hours per day. In other words, this corresponds to a differential. This change is the most important at the end of the winter and spring at high latitudes, when the sun comes back. Moreover, as explained in Sections 4 and 5, this change of insolation does not indicate only direct exposure of the air parcel to sunlight, but also mixing with air masses from lower latitudes, which were exposed to sunlight for a longer time. High values of this quantity at high latitudes can therefore also be an indicator of dynamical perturbations. This is the case in Fig.2 in late January / early February for example (as explained in Sect. 4).

8. P12 L13

PT-range, I would prefer θ-range

➢ This has been changed.

9. P13

I would recommend to write out the abbreviations MLT and ES because they are only sporadically used.

➢ The abbreviation MLT has been written out. However, we have decided to keep the abbreviation ES for the sake of readability, because it is often used together with SSW (SSW-ES).

10. P13 L31

"using the off-line wind driven isentropic transport" - I think your transport is 3d (diabatic heating), and this effect is important for the here considered time periods of the order of few months.

➢ We have changed the text to "We applied a data assimilation approach based on an extended version of the off-line wind driven isentropic transport and assimilation model DIAMOND, in which cross-isentropic transport was implemented using diabatic heating rates".

11. P15, L5-15

Here, once again Dobson units defining the contribution to the total column would be better.

➢ Please see our reply to the second comment.

12. P16, L3

This loss begins at higher altitudes in late January.

➢ This has been changed.

[revised manuscript text omitted]